# Environmental Interventions for Physical and Mental Health: Challenges and Opportunities for Greater Los Angeles

**DOI:** 10.3390/ijerph16122180

**Published:** 2019-06-20

**Authors:** Joshua F. Ceñido, C. Freeman, Shahrzad Bazargan-Hejazi

**Affiliations:** 1College of Medicine, Charles R. Drew University of Medicine and Science, 1731 E 120th Street, Los Angeles, CA 90059, USA; shahrzadbazargan@cdrewu.edu; 2Los Angeles County Department of Mental Health, 550 S Vermont Avenue, Los Angeles, CA 90020, USA; 3Los Angeles County Medical Association, 1055 W 7th Street, Suite 2290, Los Angeles, CA 90017, USA; cfreemanmdmba@hotmail.com; 4David Geffen School of Medicine, University of California, Los Angeles, 10833 Le Conte Avenue, Los Angeles, CA 90095, USA

**Keywords:** active living, built environment, health equity, health outcomes, mental health, neighborhood, urban design, walkability

## Abstract

The fields of urban planning and public health were conceived under the same pressures and goals at their inception in the 17th and 18th centuries and continue to address the health concerns of an ever-increasing urban population. While the mutual need that both philosophies have for each other becomes more tangible through research and practice, the application of their interrelatedness continues to benefit residents and visitors of mindfully-built environments. In health-conscious Los Angeles, there lacks a comprehensive assessment of health-centered considerations being implemented by those entrusted with the responsibility of shaping our cities. As a greater majority of the world’s population moves into urban settings, built environment interventions play a progressively vital role in addressing physical and mental health concerns. This piece hopes to bring to attention the need for focused and dynamic approaches in addressing health concerns by means of design, planning, and policy, by focusing on the challenges and opportunities faced by the geographic and human resources of the Greater Los Angeles area.

## 1. A Human Environment

The assurance of food security provided by the First Agricultural Revolution and its subsequent allowance for the exchange of food and ideas, solidified the impetus for high-density human settlements [1]. Improvements in the amount of food provided per square foot of land meant larger populations could be fed in a relatively small area, entailing that not all individuals were required to take on an occupation of hunting, gathering, or farming [2]. Anthropological theories tend to accredit the formation of high-density settlements to the natural economic viability of a centralized location where food could be transported, stored, and traded with a breadth of services provided by individuals who have pursued occupational specialties. The maintenance of this economic viability relies on the positive surplus of these and other benefits over liabilities such as crime and disease.

Whatever the rationale was for our prehistoric ancestors to practice waste management, the separation of unneeded and potentially harmful refuse from daily living spaces has conferred benefits on the security and evolution of societies. As early as Neolithic times, humans were found to have amassed and separated materials such as mollusk shells, animal bones, debitage, and feces in well-defined dumpsites [3]. The era appears to have marked our ancestors’ first traceable practice of functional space allocation and designation. The first humans to hone this practice as a conscious process were in theory the first designers and planners. Ancient high-density settlements, hereafter referred to as ‘cities’, were a direct result of an evolution of those conscious processes. Cities across Africa, Eurasia, and the Americas were constructed with varying systems for aqueducts and sewers to support the population’s needs for water and sanitation. The summation of these practices continues today, apart from earlier understandings of religious or social taboo, as a component necessary to both public health and urban planning.

## 2. A Healthy Environment 

Leading health concerns in recent centuries have changed as a direct result of how humans have approached the idea of their occupying a small space together. Areas with higher population densities were centers of commerce, governance, and other forms of power, and were known foci or catalysts for morbidity. Activities commonly initiated from centers of power, particularly trade and war, expedited the bubonic plague’s spread into Europe and smallpox’s introduction to Mesoamerica and Southeast Asia [4,5,6]. While the etiologies were later fully realized, it was known far beforehand that individuals were exceptionally more likely to contract infectious diseases such as cholera in industrialized cities like London, Liverpool, Chicago, and New York in comparison to the countryside and less-connected, less-industrialized settlements. Investigations, like those by Valentine Seaman addressing yellow fever in 1795 New York [7,8] and John Snow in 1854 London addressing cholera [9,10], focused on the nature of disease in its geographical context and established the importance of the built environment in disease incidence. The 18th century had already seen several coastal cities essential to regional and transoceanic trade adopting criteria and protocols for isolating the sick and establishing quarantine at the governmental level [11]. In these formative years of the practice of public health, it was realized that disease management could be approached with planned and coordinated community action.

Developed countries where public health initiatives first took hold, appear to be the first and most-effective at lowering rates of mortality attributed to communicable and infectious disease [11,12,13]. Where it was once infectious phenomena like tuberculosis, pneumonia, diarrhea, and enteritis that claimed the most American lives, it is now non-communicable diseases that have become the leading causes of death. These conditions, including ischemic heart disease, stroke, Alzheimer’s, respiratory cancers, and diabetes were once considered to be ‘diseases of affluence’ as they were historically known mainly to afflict industrialized regions with substantial urban populations [14,15]. Following the geographic spread pattern of industrialization, these conditions have become especially prevalent in the latter half of the 20th century in urban populations of developed nations and in urban populations of developing nations in the following decades. This burden of disease continues to scourge a greater proportion of individuals, regardless of social class, and are becoming widely experienced among those living below the poverty line [16,17,18].

Developments in the practices of sanitation and building code have addressed public health concerns in recent centuries, greatly improving the health conditions of city dwellers [19,20,21]. It was clear by the 1850s that overcrowding, low incomes, and death were interrelated and distinct in the urban settings of industrialized nations [22]. Initial reforms to prioritize urban sanitation and public health were accomplished with the leveraged insight that the wealthy, and thus more politically influential, were threatened by the same health risks that afflicted the poor. New York City’s squalor was a substantiated economic and political liability, mobilizing constituents from all socioeconomic classes to pressure lawmakers into imposing standards for sanitation [23]. Diverse pressures established guidelines for fire safety, access to utilities and physical accessibility, environmental control, and other necessities. It continues to take varying successions of concerted social movements to influence belief, practice, and legislation, backed by scientific rigor, to address the evolving risks of urban living [24].

Consummating efforts across disciplines bring attention to relative blinds spots historically found in respective practices. The concreteness of physical illness has fueled the advancement of interdisciplinary cooperation and aided the generation of data, indicating the need to address safety and mental health concerns, apart from the traditional realm of physical health, at an administrative level. Mental health is now acknowledged to cost the world millions of disability-adjusted life years (DALYs) and underestimated amounts of major financial loss [25,26]. Suicide has consistently ranked among the top 10 causes of death in the United States since 1975 [13,27,28]. Resulting increases in awareness of mental health concerns further generates investment and data to address these concerns. Nevertheless, public efforts that target mental illness trail behind investments in other health-related and safety issues. The World Health Organization reports neuropsychiatric illness as causing 13% of the world’s burden of disease in a 2003 report, while accounting for about only 2% of total government health budgets [29]. This discrepancy in policy and investment is associated with the current limitations in technology, deficiencies of data, and an immaturity of awareness with regards to mental illness, as compared to physical illness. Studies have yet to definitively confirm the proportion of burden of mental illness which urban centers carry when compared to rural areas. Whatever the risk to health or safety there is to be addressed, the urban setting provides an ideal opportunity for policy to deliver relief for the greatest amount of individuals.

## 3. A Humane Environment 

Urban designers and planners, policymakers, community leaders, and leaders of other specialties have worked to address the risks of living in densely populated communities. The most ambitious of these collaborations result in the tailoring of built environments to prioritize the wellbeing of inhabitants. The work of building humane environments is not novel in public health praxis. Traffic engineers and ergonomics engineers, among others, have collaborated not only to establish design standards for safer vehicles but also to create safer transportation infrastructure to address the incidence and gravity of injury on roadways [30,31,32,33], another leading cause of injury and death in the United States and the world [13,15,27,28]. Governmental organizations and designers have employed principles in their land-use and zoning practices, which improve visibility and surveillance, pedestrian traffic, and visual and emotional stimulation, all of which aim to create environments which deter crime and, in turn, aim to improve inhabitants’ perceived safety, happiness, and mental health [34,35,36,37,38]. With considerations of safe and easy access to healthy foods, entertainment, and pro-social environments, the current literature continues to provide more evidence that the physical, mental, economic, and social health of cities can be improved with deliberate design made to prioritize the manifold aspects of wellbeing.

## 4. Challenges and Opportunities for a Greater Los Angeles 

Greater Los Angeles is a major global center of cultural, economic, and political power. This metropolis is known for its global influence through media, its urban sprawl, its reliance on private vehicular transportation, and its preoccupation with health and wellbeing. With expectations to steadily continue its growth in population and to host the 2028 Summer Olympics, leaders will be entrusted with the responsibility of utilizing geography and built environments as a means for investing in and optimizing Los Angeles’s human and economic potential, particularly in the coming years. While the interrelation of physical and mental health is well-acknowledged and applied, the praxes of concerted policymaking and planning that prioritize wellbeing beyond physical health are relatively underdeveloped. Much is yet to be done to secure analogous investments in mental and social health determinants. With respect to the aforementioned, this section offers prospective matters where interventions to the built environment of Greater Los Angeles could be expected to greatly improve the quality of life for inhabitants and visitors alike.

### 4.1. Walkability

Early- and mid-20th century approaches to urbanization have prioritized design catered to the flow of motor vehicles rather than the practicality of foot traffic. Urban sprawls connected primarily by miles of roadways have established a paradigm of codependence among single-use zoning laws, low-density developments, and private vehicle use. Various practitioners of planning and design have instituted indefinite parameters, such as compactness, safeness, and attractiveness to ascertain and practice ‘walkability’ [39]. Whichever parameters are used to define walkability, its enhancement aims to promote walking as the instinctively preferred choice for traveling between locations. A review of the discourse on walkability associates walkable environments with the following notions [39]:Lively and sociable environments that are clean, pleasant, and promote social interaction among individuals from different walks of life. This is often abetted by storefronts and mixed-use developments [40];Environmentally sustainable public transportation that is accessible to individuals who are unable to use cars due to income, age, or disability;Exercise-inducing environments with shade, access to drinking fountains, places to sit, protections from vehicular traffic, and extensive street connectivity [41].

Elements which underwrite walkability encourage inhabitants to employ all methods of active travel including bicycling and public transit [42]. The importance of walkability as a health intervention continues to be acknowledged with the breadth of methods developed to describe and measure walkability. This acknowledgement in turn has strengthened the literature with evidence of its association with lower levels of environmental pollutants, less consumption of fossil fuels, increased overall physical activity and happiness, lower rates of obesity and diabetes, and better cardiovascular and mental health [43,44,45,46,47,48,49,50,51,52,53].

Drivers in Los Angeles on average experience a 45% increase in travel time due to congestion when compared to travel times free of congestion. This percentage increase in travel time is the highest in the United States and comes third only after Mexico City and Rio de Janeiro in all of the Western Hemisphere [54]. Investments made to reduce traffic, private vehicle usage, and fossil fuel reliance in Los Angeles are primed to address more than just the economic losses caused by congestion, estimated to cost Los Angeles 13.3 billion USD in 2014 [55]. This amount accounts for lost productivity, costs of transportation and fuel, but may not fully consider the effect of roadway congestion on commuter and community health outcomes. For example, county-wide studies have found adverse birth outcomes to be geographically linked to residential proximity to high-volume traffic [56,57]. Built environment interventions to improve walkability enact positive changes across an accumulation of interrelated behaviors and health outcomes. Due to the depth and complexity of the interrelation, it may be difficult to quantify a projected benefit from these interventions to improve walkability, having to consider public and consumer cost savings, land use efficiency and valuation, community livability and safety, and social equity [58,59,60]. 

In Los Angeles, improvements in infrastructure which make active transport more appealing and feasible while ameliorating the flow of roadway transportation, are expected to improve the economic and social health of communities. A 2015 county-wide study found that households in rail transit corridor neighborhoods drove less and rode public transportation more often than the county average [61]. Communities near older rail transit corridors with well-integrated, economically active, mixed-use developments had residents who, on a daily basis, drove 11 miles fewer in comparison to residents of neighborhoods in newer rail corridors. This indicates that considerations of walkability with regards to the availability of public transportation may be best optimized in the context of the mixed land use developments. Isochrone mapping of transportation duration times and mode preferences across Los Angeles, when overlaid with economic, health, and zoning data, might demonstrate notable relationships between transportation behaviors and access [62].

Microscale investigations in Pittsburgh and New York City associate economically healthy neighborhoods with increased walkability factors and increased pedestrian traffic [63]. Pedestrians were likelier to walk streets where land was being used rather than where there were empty lots, and were even likelier to walk streets where land was being occupied by more buildings for human use, rather than parcels used for other purposes such as parking. Walkability was enhanced by transparent and contiguous storefronts, well-maintained infrastructure, and street cafés. These environments were built to facilitate travel at the speed of human locomotion, which includes benches, street vendors, small signage, as well as a feeling of ‘enclosure’ where trees and buildings provide vertical elements which help define an outdoor space as manageably walkable and more comfortable [64]. Green spaces that act as enclosures along pedestrian paths add sensational complexity to urban environments, encouraging pro-social interaction, while providing positive associations with physical, mental, and subjective health [65,66,67]. These microscale studies of American cities find that the most walkable environments provide visual complexity with varying building facades, varied geometry, and displays of art, symbols, history, and culture relevant to the identity of the geography, thus making the experience of walking more interesting. Many of these elements have been well considered in the design of Los Angeles’s replacement for its Sixth Street Viaduct with its allusion to the city’s architectural history and accompanying green space. Communities with more walkability were socioeconomically advantaged over those with lower walkability scores. Strong economic viability with walkability were found where storefronts were adjacent, clean, safe, active, and transparent in spaces that provided a safe feeling of enclosure. The most successful and active footpaths were those well-connected by public transportation, especially those along transit corridors [44,63]. The finding that communities with higher walkability scores tend to have greater socioeconomic advantages strengthens the practicality of designing built environments that prioritize the pedestrian. It is thus in the interest of policymakers, planners, and designers to design and build environments which facilitate walkability.

### 4.2. Safety

Residents who do not perceive their neighborhoods to be safe are likely to be socially isolated and live sedentary lifestyles [68]. California’s Institute for Local Government posits from qualitative data that the choice to minimize time spent on neighborhood streets is directed by the implicit purpose of harm reduction [69]. In the context of built environment interventions, harm reduction involves facilitating security from physical accidents and from crime. This section aims to address how potential pedestrians perceive their vulnerability to these dangers impact health outcomes, and how built environment interventions may curtail related concerns.

When speaking on the subject of their children walking to school, some parents in Los Angeles prefer that their children avoid roads with high vehicular and commercial activity, citing concerns for physical safety around vehicular traffic [70]. This influenced parents to instead drive their children to and from school and across otherwise walkable distances, and at times discourage walking as a form of travel altogether. For many roads in Los Angeles, a commercial or human presence denotes higher vehicular traffic volume and speeds. These roads were designed with the belief that larger roads and lanes prevented traffic accidents, not considering that these wider lanes encouraged drivers to travel at higher speeds, which in turn lower pedestrian safety [71,72]. There is well-documented evidence now that the severity of individual pedestrian injuries from vehicular impact is worse in low-density residential areas when compared to downtown and compact residential areas where more foot and vehicular traffic might be expected [73,74]. These compact, mixed-use downtown areas have narrower auto lanes, bicycle lanes, uneven road surfaces, and trees which envelope roadways and obstruct intersection visibility for drivers. These road design elements are hypothesized to increase the perception of risk in drivers, communicating more so than a clear, wide, and smooth path that there is less room for driver error [75]. Consequent driving speed reductions and greater driver caution have made narrower roadways objectively and subjectively safer for all roadway users [74,75,76]. Vertical elements and complexity appear to invite a human presence at a controlled pace, welcoming the pedestrian and slowing the driver. Parents and pedestrians in Los Angeles may prefer a broader application of these elements to amend older designs that cater to the overconfidence of drivers.

While some understand vehicular danger to be the primary safety concern for pedestrians, children in inner city Los Angeles neighborhoods express a preference to walk along major roads that tend towards high vehicular traffic volume, holding the perception that busier roads feel safer than quieter residential roads [70]. Further investigation across Southern California communities found that those who prefer to walk along busier roads, which may proffer greater vehicular danger, avoid smaller and quieter roads due to perceived associations with crime, drugs, and gang activity [70,77]. Children tend to prefer streets with more commercial activity that offer more opportunities for social interaction and safety in the form of surveillance [77]. These sentiments are supported by various studies of differing age groups and in turn support Jane Jacobs’s ‘eyes on the street’ concept [78,79], purporting that the creation of pro-social public spaces supports walkability by establishing a built environment which deters criminal activity and promotes perceived safety. Individuals who live in pro-social, walkable areas that are thought of as less crime-prone are found to have lower levels of psychological distress, lower rates of obesity, and lower rates of reported chronic illness [79,80]. This perceived safety in social support is reinforced by environments designed so that others can easily maintain visibility of potentially unsafe areas, thereby discouraging the anticipation of and actual occurrence of criminal activity [68]. Trees serve as a narrow, columnar vertical design element which, while also providing comfort and complexity, affords ample visibility to pedestrians and others traveling at slower speeds [64,65,66,67,74,80].

Navigability is another key component of perceived safety, as individuals are more likely to explore and use a space if they feel they know where they are and know how to find their way to other destinations. Navigability is assisted by well-connected paths and well-lit, well-labeled signs and directional information, affording travelers confidence in their ability to explore a space [81]. Communities with greater navigability host individuals who traveled farther and more often, use a greater number of unique pathways, and utilize more neighborhood services [82]. The presence of landmarks not only invites a human presence, but provides a unique identity to an environment and a point of orientation for travelers, fostering neighborhood navigability [64,83]. Just as the Christ the Redeemer Statue, Eiffel Tower, Empire State Building, and Namsan Tower serve their cities as points of orientation and symbols of identity, so might largescale construction feats in Los Angeles’s future serve its urban sprawl. Already, landmarks like Los Angeles City Hall, the Downtown Los Angeles skyline, Santa Monica Pier, and the Watts Towers serve as symbols of identity and points of orientation for their respective localities. 

Pro-social environmental factors such as navigability, visibility and transparency, cleanliness, and built elements such as street furniture, bicycle lanes, crosswalks, storefronts, and eateries confer a maximization of human presence to a place, which further enhances the perception of safety and walkability. Perceived safety and walkability have a greater influence on behavior and health outcomes when compared to metrics of objective, physical walkability [84], making the argument that the apparent attractiveness of a place or route is contributory to its utilitarian functionality. To encourage active transport behaviors, improve health outcomes, and support local economies, it is not enough that Angelenos are promised a safe environment; it is essential to the appeal of health-positive environments that places feel safe. While the improvement of objective safety metrics prevents immediate harm and fatality, how perceived safety invites individuals to utilize a space prevents adverse downstream health outcomes.

### 4.3. Resource Security

Americans who experience housing instability and food insecurity often choose to forgo necessary health care and medications in order to make other ends meet. These individuals are more likely to have poor access to healthcare and therefore suffer greater health burdens personally and financially [85] because they are more likely to seek medical care through emergency services, when the course of their illness is advanced and would likely require more time spent in the hospital. If economic incentives are to be a primary driving force behind investing in the creation of pedestrian-friendly communities, considerations must be made to ensure that residents in lower-income neighborhoods, where these built environment interventions are expected to provide the greatest positive change, are not excluded from their communities as the economic power of their spatial geography increases.

Socioeconomic pressures have influenced the health and demographic landscape of Greater Los Angeles and other metropolises for better and worse. A study conducted from 1998–2008, found areas of Los Angeles that had suffered from inadequate provisions of social services due to socioeconomic pressures on provider facilities [86]. Property values have risen following increases in walkability and economic activity in areas like Downtown Los Angeles, Hollywood, Santa Monica, and Venice. As a result, social service providers were priced out of the community, unable to move to another part of the same neighborhood or even expand operations within the community [86,87]. Gentrification and the increasing presence of higher-income residents strengthened local grassroots opposition to the establishment of new social service facilities, particularly for subsidized housing and mental health provisions [86]. Individuals who were already under-resourced faced these new stressors and barriers to accessing services involving health, food, housing, employment, substance abuse treatment, and advocacy. Some residents were being priced out of their homes and communities altogether. In cases where a physical environment is made conducive to good health, the increased desirability of a neighborhood’s real estate aggravates socioeconomic and health disparities.

Already afflicted by comparative economic stagnation, higher crime rates, and poor walkability, neighborhoods across South and Central Los Angeles are disproportionately burdened as well by poor food security. Surveys from 2004–2006 of three primarily Latino neighborhoods, with poverty rates three times the national average, reported only 2% of retail food outlets to be supermarkets, while fast-food restaurants and convenience/liquor stores accounted for 30% and 22% of retail food outlets respectively [88]. Primarily African American neighborhoods in South Los Angeles were found to have significantly less healthy food options in their vicinities when compared to more affluent neighborhoods in Los Angeles County [89]. Since Latino and African Americans in Los Angeles are significantly more likely to suffer from obesity, diabetes, and cardiovascular disease when compared to Asian and white Americans, and because the health burden of these chronic diseases fall more heavily upon lower-income communities [90], an array of opportunities exist to employ built environment interventions to zoning and accessibility as a tool for social justice in health equity.

Thoughtful urban agriculture may be a valuable device for health equity. Of those benefits known to promote physical activity and active transportation, urban agriculture in the modern city presents opportunities for food access, land utilization, and green infrastructure. Originally a strategy for financial and food security in developing nations, urban agricultural projects in large American cities are advertised alongside social justice issues, though they have yielded inconsistent and concerning results [91,92]. In recent decades, urban agricultural and other environmental justice projects in San Francisco and New York City, touting the guise of sustainability, have become harbingers of gentrification to their communities. The benefits of these projects were essentially appropriated by high-end developers who catalyzed the displacement of lower-income residents from their homes [93,94]. In the context of socioeconomic and health disparities, neighborhoods with poor food security require more from urban agriculture projects, beyond the provision of affordable, healthy food. These interventions cannot be considered successful if resultant pressures force the already under-resourced locals from their homes.

The injustice of creating a seemingly healthier built environment without protecting the community’s ownership of their neighborhood’s potential is contradictory to the virtue of health-conscious built environment interventions and the principle of non-maleficence. Throughout American history, communities of color who often worked the land were prevented from owning and managing land [92]. The 14-acre South Central Farm in Los Angeles was a self-organized community endeavor formed by 360 families primarily of indigenous Mesoamerican descent. This urban farm provided residents with a sense of community, accountability, and ownership of the choice, distribution, and management of its products, which consisted of over 100 plant species relevant to the population [95,96]. Politico-economic pressures brought an end to the farm, forcing its closure in 2006 despite extensive community protest and the farm’s proven viability as an asset to community health, culture, identity, and autonomy. City leaders and developers of the time thought the land to be more valuable as an industrial or commercial asset and failed to ascertain the downstream costs of their actions. While some of the land was offered back to farmers years later, many had already left the community. Some others pursued food and environmental justice endeavors with goals such as promoting food sovereignty and establishing green infrastructure for the community [97,98].

The socioeconomic milieu that guides the various forces behind community displacement is complex and beyond the scope of this piece. The purpose of acknowledging residential displacement and resource insecurity as consequences of health-positive built environment interventions is to provide a broader assessment of potential complications already observed in practice in Los Angeles and elsewhere. Of particular concern are the fates of those with mental illness, who suffer greatly from resource insecurity and intrinsic barriers to personal agency. Compromised agency in the setting of societal stigmatization has precluded many of those with mental illness from receiving critical attention across many disciplines, underscoring the importance of mental health awareness and support. Literature on the subject of environmental intervention through planning and policy in relation to mental health is fledgling but growing. California adopted a Housing First model in 2016, supporting Los Angeles’s goal to house more than 40,000 homeless individuals [99,100]. This model follows evidence showing that chronically homeless individuals, particularly those with mental illness, who are then established in stable housing incur less medical costs and experience improved health outcomes [100]. Conversely, a 2017 study found that those displaced from their longtime homes in gentrifying communities were significantly more likely to suffer from mental illness, to make emergency department visits, and to be hospitalized when compared to residents who were able to keep their home [101]. These individuals also suffered from malnutrition, heart disease, substance abuse, and other chronic illnesses. Despite the relative paucity of research literature on the relationship between built environment interventions and mental illness, there is developing interest in the potential that housing security has to improve mental and physical health outcomes.

The cost of living in any geography can be expected to rise as the quality of life improves. In that regard, thoroughly fostering social justice in health equity requires leaders to act to avert the displacement of residents from their homes and ensure that health-conscious changes to the neighborhood prioritize its residents [102]. In order to create an ideal metropolis, enlightened civic leaders will be expected to make exhaustive considerations for their jurisdictions and to look past face-value short-term economic gains. Whether or not financial benefit should be a chief objective, the appeal of economic incentives can be an effective tool for prioritizing public wellbeing. Discussions on the topic of environmental interventions for health in terms of financial cost-benefit analyses may expediently communicate the discussion’s significance to a wider audience. Since the concept of cost is universal to all disciplines, these considerations may compel leaders from across disciplines to rise to the responsibility.

Community, government, business, academic, and thought leaders have influence over the health of Angelenos beyond the delivery of secondary and tertiary preventative healthcare services. Primary prevention of disease demands investment and wisdom directed towards the alterable determinants of health [103]. In the formation of neighborhoods with ideal living environments, the requisite reminder is that safety, sustainability, and access cannot be high-end commodities if health equity and optimization are to be achieved.

## 5. How to Build a Healthier Greater Los Angeles 

Ushers of forthcoming ventures borne in Los Angeles will learn from and improve upon examples set in other times and places. Enterprising Angelenos will explore, adapt, and transform this and other metropolises facing barriers to resource security and other health factors. Public health and urban planning leaders have faced many challenges unique to the urban setting, allowing modern populations to take several groundbreaking systems and outcomes for granted. As a result, Los Angeles in general has reliable waste management, sewage, utilities, and roadway systems. Some frameworks founded elsewhere exist to address ongoing health and urban design concerns, and can be more widely applied or incentivized, such as the Leadership in Energy and Environmental Design (LEED) criteria and certification established by the United States Green Building Council. Multi-jurisdictional and public-private feats comprise evidence of the faculty that cooperation can rally in the interest of wellbeing; these are programs like the Los Angeles City’s Urban Agriculture Incentive Zone Ordinance, public transit discounts from the Los Angeles County Metropolitan Transportation Authority for students and seniors, allowances for low-emission vehicle utilization of high occupancy vehicle (HOV) lanes, the adoption of the Housing First model, and the Sustainable City pLAn. The complexity and virtue of mobilizing human and environmental resources to optimize a people’s health justify the labors of this ongoing pursuit.

## 6. Conclusions

In the formation of the greatest and most enduring settlements, inhabitants have innovated and found solutions to improve and enrich living conditions. An ever-increasing complexity of interrelated challenges faced by modern populations necessitates multidisciplinary, multidimensional approaches to maximize the health, safety, and wellbeing of any community. There is little precedent for a dynamic metropolis as geographically and administratively decentralized as Los Angeles, to follow to that end. Inclusive community participation in planning for Greater Los Angeles requires the voice and presence of many knowledgeable, engaged individuals, representing more than a hundred overlapping neighborhoods and independent city governments. Thoughtful planning for Greater Los Angeles requires keen and sustained exploration of this interdisciplinary convolution, providing context for community, government, economic, planning, design, and health leaders to operate. Once clear on the agenda, the opportunities and innovation which derive from this cooperation will offer provisions for a healthier and more humane Los Angeles.

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
