# Peer review of "Environmental Interventions for Physical and Mental Health: Challenges and Opportunities for Greater Los Angeles"

_ijerph, 2019, doi:10.3390/ijerph16122180_

Round 1

Reviewer 1 Report

It is a well-intentioned work on the relationship between urban planning and health. An exhaustive literature review has been made and some interesting ideas are pointed out such as the relationship between planning and social justice. However, it has two weaknesses to improve:

1. Does not delve into the relationship existing between the actions to improve urban space and its gentrification (except one sentence)

2. Does not concrete the actions that could be carried out in Los Angeles to get a healthier space. The text is very generic in this regard. There is no concrete allusion to urban planning that is developing in the city.

Author Response

Comments and Suggestions for Authors

It is a well-intentioned work on the relationship between urban planning and health. An exhaustive literature review has been made and some interesting ideas are pointed out such as the relationship between planning and social justice. However, it has two weaknesses to improve:

1. Does not delve into the relationship existing between the actions to improve urban space and its gentrification (except one sentence)

Authors’ Response:

The authors acknowledge the significance and complexity of this relationship. We have worked to elaborate further on the topic, particularly in subsection 4.3, though we acknowledge that we will not be able to fully attend to all significant aspects of gentrification as it relates to shaping urban spaces for better health equity. We hope to highlight the importance of these considerations, and to provide context related to Los Angeles and other American metropolises. We thank you for this opportunity to elaborate on the issue further.

2. Does not concrete the actions that could be carried out in Los Angeles to get a healthier space. The text is very generic in this regard. There is no concrete allusion to urban planning that is developing in the city.

Authors’ Response:

We have taken the opportunity to elaborate on the failures and anticipated successes of projects across Los Angeles to accomplish a more health-positive environment. This includes references to the Sixth Street Viaduct (line 207), South Central Farmers (lines 343-354), the adoption of a Housing First model to build and provide affordable housing (lines 364, 406).

Reviewer 2 Report

This paper examines some potential environmental interventions to improve human health in cities. The authors focus on the opportunities and challenges for interventions that increase the walkability, safety and access to resources for people living in the Greater Los Angeles area.

I applaud the idea of emphasizing the shared objectives of public health and urban design and of raising awareness of the potential for economic and public health synergies when urban design improves walkability, safety and access to resources. However, I think the paper in its current form does not effectively achieve this aim. I think the article could be more effective if it highlighted the most important urban design concepts for public health (walkability, safety, resource access), as it currently does, and then applied those concept to Los Angeles as a case study. As written, it currently seems very niche and not of broader appeal, although the concepts highlighted within the paper certainly apply to many cities around the world.

Generally, the writing style is verbose and lacks clarity in several places. Several sentences are long (e.g. lines 95-99, 146-150) and several others are confusingly written, such as the following. I highlight the confusing components:

Lines 22-23: This sentence is awkward. Perhaps rephrase as "particularly in metropolises known for their sprawl and heavy..."

Line 36: change to “natural economic viability” (not naturally)

Lines 51-52: This is a confusing sentence. By expanding on the commerce/governance/power topic, leads the reader away from the point, which is to say “Cities are known foci or catalysts of disease” (also, “disease morbidity” is redundant). Please rephrase.

Lines 68-70: The current trends around non-communicable diseases cannot be simply described as “diseases of affluence”. Please rephrase akin to: “…American lives, it is now non-communicable diseases that are the leading causes of death. These conditions, including … and diabetes, have become especially prevalent since the late twentieth century in urban populations of developed countries.” Or similar.

Lines 77-81: This long sentence loses its focus. Especially the “varying successions of… urban living” part. What do the authors mean by “social movements in belief, practice, and legislation”? Perhaps clarify and cite?

Lines 82-83: “…the merits of… bring attention to…”

Lines 132-134: “Focus from… have instituted nebulous parameters… to ascertain and practice walkability”

Lines 162:  Missing “be” – “it may be difficult”

Line 186: “These case studies…” which case studies?

Line 195: “linked and further explore in this piece” Where are they further explored? This sentence comes at the end of the “Walkability” section.

Lines 195-198: Redundant. Actually, 195-198 could be cut and lines 198-200 just added to conclude the previous paragraph.

Lines 203-204: This is a generalization. Please rephrase.

Line 216: could you clarify what you mean by “severity of pedestrian injury from…”? Do you mean the severity of the injuries or the prevalence of injury (severity of the problem)?

Line 256: Do you mean navigability here or walkability (the mention of bike lanes suggests navigability)?

Line 265: Change “insecurity often to choose” to “often also” or “often too”

Line 268: “…burdens both to their person and financially” change to “burdens personally and financially”

Lines 279-281: “…were priced out of… moving to another part of the same neighbourhood…”

Author Response

This paper examines some potential environmental interventions to improve human health in cities. The authors focus on the opportunities and challenges for interventions that increase the walkability, safety and access to resources for people living in the Greater Los Angeles area.

I applaud the idea of emphasizing the shared objectives of public health and urban design and of raising awareness of the potential for economic and public health synergies when urban design improves walkability, safety and access to resources. However, I think the paper in its current form does not effectively achieve this aim. I think the article could be more effective if it highlighted the most important urban design concepts for public health (walkability, safety, resource access), as it currently does, and then applied those concept to Los Angeles as a case study. As written, it currently seems very niche and not of broader appeal, although the concepts highlighted within the paper certainly apply to many cities around the world.

Generally, the writing style is verbose and lacks clarity in several places. Several sentences are long (e.g. lines 95-99, 146-150) and several others are confusingly written, such as the following. I highlight the confusing components:

Authors’ Response:

The authors thank you sincerely for this feedback.  Changes have been made throughout the piece, including to those lines/sentences you have brought to our attention.  These and all alterations aim for more concise and direct language.  Section 5 and subsection 4.3 in particular were expanded to highlight how concerns and interventions may be addressed in Los Angeles.  We also provide in subsection 4.3 a short instance of relevant/recent historical context for Los Angeles (lines 343-354).  In section 5, we then highlighted what Angelenos have done and might have to do to confront the environmental obstacles to wellbeing.

Lines 22-23: This sentence is awkward. Perhaps rephrase as "particularly in metropolises known for their sprawl and heavy..."

This text was removed in revisions.

Line 36: change to “natural economic viability” (not naturally)

This is duly noted and the sentence was changed accordingly.

Lines 51-52: This is a confusing sentence. By expanding on the commerce/governance/power topic, leads the reader away from the point, which is to say “Cities are known foci or catalysts of disease” (also, “disease morbidity” is redundant). Please rephrase.

This sentence was changed to “Areas with higher population densities were centers of commerce, governance, and other forms of power, and were known foci or catalysts for morbidity.” (now line 59) The sentence that follows was altered to highlight the relevance of ‘centers of power’.  This is further highlighted at the beginning of Section 4.

Lines 68-70: The current trends around non-communicable diseases cannot be simply described as “diseases of affluence”. Please rephrase akin to: “…American lives, it is now non-communicable diseases that are the leading causes of death. These conditions, including … and diabetes, have become especially prevalent since the late twentieth century in urban populations of developed countries.” Or similar.

The structure of the sentence and paragraph was amended with considerations of this suggestion and with the purpose of clarification. (Lines 76-83) “These conditions, including ischemic heart disease, stroke, Alzheimer’s, respiratory cancers, and diabetes, were once considered to be ‘diseases of affluence’, as they were historically known mainly to afflict industrialized regions with substantial urban populations.”

Lines 77-81: This long sentence loses its focus. Especially the “varying successions of… urban living” part. What do the authors mean by “social movements in belief, practice, and legislation”? Perhaps clarify and cite?

We thank you for this feedback.  We have improved the language to better communicate our original idea. (this text is now lines 93-95).  The authors were able to elaborate and discuss this idea in sections 4 and 5 of the manuscript. .

Lines 82-83: “…the merits of… bring attention to…”

The sentence was edited for conciseness and clearer communication.

Lines 132-134: “Focus from… have instituted nebulous parameters… to ascertain and practice walkability”

The sentence was edited for conciseness and clearer communication.

Lines 162:  Missing “be” – “it may be difficult”

Thank you. We have made the correction.

Line 186: “These case studies…” which case studies?
These case studies are those first referred to at the beginning of the paragraph: “Microscale investigations in Pittsburgh and New York City..” We altered the text to better communicate this.(lines 203-204)

Line 195: “linked and further explore in this piece” Where are they further explored? This sentence comes at the end of the “Walkability” section.

Referring to the feedback comment item immediately below, the authors feel that the resulting edit required that we remove the sentence from the text.

Lines 195-198: Redundant. Actually, 195-198 could be cut and lines 198-200 just added to conclude the previous paragraph.

This is duly noted and the text was changed accordingly.

Lines 203-204: This is a generalization. Please rephrase.

This is duly noted and the sentence was replaced for better clarity.

Line 216: could you clarify what you mean by “severity of pedestrian injury from…”? Do you mean the severity of the injuries or the prevalence of injury (severity of the problem)?

We mean for ‘severity’ to refer to how damaging or acute or grave individual injuries may be. We have altered this to say “…the severity of individual pedestrian injuries from….”(line 233)

Line 256: Do you mean navigability here or walkability (the mention of bike lanes suggests navigability)?

While the presence of bike lanes may confer a quality of navigability, we mean to refer to ‘walkability’ section (4.1) where it is noted that the presence of bike lanes indeed contributes to the quality of walkability.

Line 265: Change “insecurity often to choose” to “often also” or “often too”

We have edited the sentence accordingly.

Line 268: “…burdens both to their person and financially” change to “burdens personally and financially”

This is duly noted and the sentence was changed accordingly.

Lines 279-281: “…were priced out of… moving to another part of the same neighbourhood…”

Thank you for this feedback; we have edited the sentence for better clarity. (Lines 303-304) “As a result, social service providers were priced out of the community, unable to move to another part of the same neighborhood or even expand operations within the community.”

Reviewer 3 Report

In this work the authors provide a history of the evolution of human settlements in the context of the influence of the built environment on community health and well-being. From there, the authors discuss challenges and potential opportunities for regulating urban planning in the Los Angeles area moving forward. Most emphasis is placed on the need to improve walkability, safety and resource security within neighborhoods. This commentary is well written, includes extensive research and would be of interest to readers of IJEPRH. However, there are several shortcomings within the paper which should be addressed.

Major problems:

There is a disconnect between what the title and abstract suggest what the paper is about compared to the content of the paper. For example, the abstract indicates that this paper will outline focused approaches to urban planning in the greater Los Angeles area, but then the paper begins with a discussion of food security and waste management in human settlements starting in ancient times which is not particularly germane to the main topic at hand. Furthermore, the issues discussed in the paper, including amelioration of traffic congestion and development of better mixed-use neighborhoods, are relevant to any major city in the U.S. and are not specific to the Los Angeles area. The paper should be edited to follow a stronger central thesis throughout.

More attention to detail should be placed on the references section. Many journal references are not cited properly (e.g. 17, 84), doctoral dissertations should include a university and location (e.g. 33,34), and websites should include a data of accession (e.g. 52, 65, 77) or web address (e.g. 12, 27).  Reference 53 is troubling since this is a news website which cites a non-peer-reviewed study from another website, and the methodology for this study is not readily available. Therefore it is difficult to discern how traffic congestion causes costs $19.2 billion annually in Los Angeles.

Minor problems:

Lines 31-33: provide a reference

Lines 66-80: paragraph should be split into 2: (diseases of affluence (66-75) and sanitation (76-81)

Line 117: change the word ‘essential’ to another adjective that is less paramount, such as ‘significant’ or ‘important’.  Essential, like essential nutrients, suggests that the global network cannot survive without it.

Line 193: change ‘long’ to ‘along’

Lines 195-198: reword this sentence for clarity

Line 209: change ‘parents’ to ‘some parents’

Lines 209-223: It is not entirely clear from this paragraph as to whether pedestrians should be encouraged to walk in high-density areas or low-density areas. High density mixed use areas have lower potential severity of pedestrian injury but include narrower lanes, bicycles, uneven road surfaces and trees which obstruct vision.

Line 233: change Jacobs’s to Jacobs’

Line 265: change ‘often to choose’ to ‘often choose’

Line 276: change ‘finds’ to ‘found’

Lines 278-281: run-on sentence

Lines 287-299: How do you address the lack of healthy food options in some neighborhoods? Currently the market dictates which businesses are profitable in a given area which in this case leads to a higher proportion of fast food restaurants and convenience stores. Should municipal governments step in to give tax incentives for grocery stores to re-locate or offer land for open air markets?

Line 311: delete the comma

Line 313: change ‘economy’ to ‘economic’

Line 314: change ‘ensue this’ to ‘ensue in this’

Author Response

In this work the authors provide a history of the evolution of human settlements in the context of the influence of the built environment on community health and well-being. From there, the authors discuss challenges and potential opportunities for regulating urban planning in the Los Angeles area moving forward. Most emphasis is placed on the need to improve walkability, safety and resource security within neighborhoods. This commentary is well written, includes extensive research and would be of interest to readers of IJEPRH. However, there are several shortcomings within the paper which should be addressed.

Major problems:

There is a disconnect between what the title and abstract suggest what the paper is about compared to the content of the paper. For example, the abstract indicates that this paper will outline focused approaches to urban planning in the greater Los Angeles area, but then the paper begins with a discussion of food security and waste management in human settlements starting in ancient times which is not particularly germane to the main topic at hand. Furthermore, the issues discussed in the paper, including amelioration of traffic congestion and development of better mixed-use neighborhoods, are relevant to any major city in the U.S. and are not specific to the Los Angeles area. The paper should be edited to follow a stronger central thesis throughout.

 Authors Response:

Thank you sincerely for this feedback. Changes and additions have been made throughout the entire manuscript to better address the needs and opportunities of Los Angeles in particular, with an expanded highlight in section 5. A minor change to the abstract was made to introduce and summarize the topics addressed in our manuscript more accurately. The authors may have failed to communicate the importance of providing an evolution of how space allocation became a conscious process that meets the interests of multiple disciplines with the shared goal of improved health outcomes and quality of life. We hope that our alterations communicate these ideas more clearly throughout the paper.

More attention to detail should be placed on the references section. Many journal references are not cited properly (e.g. 17, 84), doctoral dissertations should include a university and location (e.g. 33,34), and websites should include a data of accession (e.g. 52, 65, 77) or web address (e.g. 12, 27).  Reference 53 is troubling since this is a news website which cites a non-peer-reviewed study from another website, and the methodology for this study is not readily available. Therefore it is difficult to discern how traffic congestion causes costs $19.2 billion annually in Los Angeles.

 Authors Response

We have made major alterations to the reference section, answering as well to each of the items mentioned above. Accession dates were provided and the section’s layout was refined.

Regarding Reference 53 (now reference 55), I have replaced this with a report released by Texas A&M Transportation Institute in 2014 and included appropriate data from the report in the body of the text.

Minor problems:

Authors Response:

Lines 31-33: provide a reference

Reference 1 is provided. 

Lines 66-80: paragraph should be split into 2: (diseases of affluence (66-75) and sanitation (76-81)

Changes were made to address the suggestion, including some clarification on the two topics. (lines 72-95)

Line 117: change the word ‘essential’ to another adjective that is less paramount, such as ‘significant’ or ‘important’.  Essential, like essential nutrients, suggests that the global network cannot survive without it.

Changes were made to address the suggestion with a simplification of language. (Line 131)

Line 193: change ‘long’ to ‘along’

This is duly noted and the sentence was changed accordingly.

Lines 195-198: reword this sentence for clarity

 Changes were made to address the suggestion, including some clarification on the two topics. (213-215)

Line 209: change ‘parents’ to ‘some parents’

This is duly noted and the sentence was changed accordingly.

Lines 209-223: It is not entirely clear from this paragraph as to whether pedestrians should be encouraged to walk in high-density areas or low-density areas. High density mixed use areas have lower potential severity of pedestrian injury but include narrower lanes, bicycles, uneven road surfaces and trees which obstruct vision.

Changes were made to the paragraph and throughout the paper to more clearly communicate the idea. Section 4.2 discusses an old design paradigm which prioritizes driver safety with large lanes which encouraged driver carelessness. This was replaced with the idea that high-density areas lead drivers to perceive more danger, causing them to slow down; these reduced driving speeds in turn were safer for pedestrians (despite the perceived dangers of high-density areas). Perceived dangers applied only to drivers in high-density areas, causing them to slow down.

Line 233: change Jacobs’s to Jacobs’

After some consultation, we believe Jacobs’s is the appropriate choice.

Line 265: change ‘often to choose’ to ‘often choose’

This is duly noted and the sentence was changed accordingly.

Line 276: change ‘finds’ to ‘found’

This is duly noted and the sentence was changed accordingly.

Lines 278-281: run-on sentence

 Changes were made for better clarification and to address the suggestion. (lines 307-310)

Lines 287-299: How do you address the lack of healthy food options in some neighborhoods? Currently the market dictates which businesses are profitable in a given area which in this case leads to a higher proportion of fast food restaurants and convenience stores. Should municipal governments step in to give tax incentives for grocery stores to re-locate or offer land for open air markets?

The authors thank you for this input. We have altered and added to subsection 4.3 so as to provide context for this issue as it pertains to Los Angeles and other cities. The text now elaborates on how incentives and cooperation from different disciplines and jurisdictions have worked together in the past towards the goal of improving the health-promoting quality of Los Angeles specifically and of cities in general. Section 5 ends with a call to action in the contexts of actions taken in the past.

Line 311: delete the comma

The text was changed accordingly.

Line 313: change ‘economy’ to ‘economic’

The text was changed accordingly. 

Line 314: change ‘ensue this’ to ‘ensue in this’

 This error and confusion was noted and the word choice was changed for improved clarity.

Round 2

Reviewer 2 Report

I believe the authors have sufficiently addressed my and the other reviewers comments and the improved manuscript is ready for publication.